# Leaf ethanolic extract of *Etlingera hemesphaerica* Blume mitigates defects in fetal anatomy and endochondral ossification due to mercuric chloride during the post-implantation period in *Mus musculus*

Aceng Ruyani[1,2]*, Eda Kartika[1], Deni Parlindungan[3], Riza Julian Putra[5], Agus Sundaryono[1,4], Agus Susanta[1]

1 Graduate School of Science Education, Bengkulu University, Bengkulu, Indonesia, 2 Department of Biology Education, Bengkulu University, Bengkulu, Indonesia, 3 Department of Science Education, Bengkulu University, Bengkulu, Indonesia, 4 Department of Chemistry Education, Bengkulu University, Bengkulu, Indonesia, 5 Department of Biology Education, State Islamic Institute of Kerinci, Sungai Penuh, Jambi, Indonesia

* ruyani@unib.ac.id

## Abstract

This study aimed to investigate the effectiveness of leaf ethanolic extract of *Etlingera hemisphaerica* (LE3H) in reducing defects in fetal anatomy and endochondral ossification in mice induced by $HgCl_2$ during the post-implantation period. Pregnant mice were divided into four groups, each consisting of 10 dams, and received drink and food *ad libitum*. The first group was administered LE3H (E1), the second one $HgCl_2$ (E2), the third one $HgCl_2$+LE3H (E3), and the fourth was control (E0), administered double-distilled water only. $HgCl_2$ (5 mg/kg bw) was administrated by injection intraperitoneally on gestation day (GD)9 and LE3H (0.39 mg/g bw) was administered by gavage on GD10. The treated and control animals were killed by cervical dislocation on GD18, dissected, and the morphologically normal living fetuses (MNLF) were collected. The MNLF of E0, E1, E2, and E3 from 5 dams were fixed with Bouin solution, and observed using the free hand razor blade technique for soft tissue examination. The remaining MNLF were fixed with 96% ethanol, and then stained with Alizarin Red S and Alcian Blue for ossification examination. Index of length of ossified part (ILOP) of humerus, index of width of ossified part (IWOP) of humerus, ILOP of femur, and IWOP of femur were calculated. E2 had higher cases of anatomical defects (74,6%) than E3 (48.9%), E1 (15.0%), and E0 (0%). E2 had humerus IWOP of 0.82±0.03, which was significantly lower than that of E0 (0.89±0.04) and E1 (0.89±0.03), while that of E1 and E0 was not significantly different from each other. Meanwhile, IWOP in E3 (0.88±0.03) was significantly higher than that in E2, but not different from that in E1 and E0. Thus, LE3H mitigated defects in fetal anatomy and endochondral ossification induced by $HgCl_2$ in mice.

**Data Availability Statement:** All relevant data are within the manuscript and its Supporting Information files.

**Funding:** This research was funded by Directorate of Higher education, Ministry of Education and Culture, Republic of Indonesia.

**Competing interests:** The authors declare no competing financial interests.

# 1. Introduction

Both developing and developed countries have similar possibilities of mercury (Hg) poisoning accidents. In North Carolina, USA, it was reported that out of 221 pregnant women, 63.8% and 100% had contamination of Hg and lead (Pb) in their blood respectively due to the use of herbicides, cosmetics, and electronic equipment [1]. Meanwhile, the Hg poisoning in Bengkulu, Indonesia, is caused by a low understanding about the dangers of Hg, the lack of effective regulation by a governmental authority, and mainly the involvement in gold mining activities [2]. Hg is widely recognized as a major environmental pollutant and has been shown to be teratogenic in humans and animals [3]. Human fatal dose of mercuric chloride ($HgCl_2$) is 1–4 g, and $HgCl_2$ can produce various toxic effects such as corrosive injury, gastrointestinal disturbances, acute renal failure, circulatory collapse, and finally death [4]. Hg and its compounds possess strong neurotoxicity that can damage the central and peripheral nervous systems [5]. It is important that people be aware of the potential toxicity of $HgCl_2$, and it is strongly recommended that a close observation and aggressive supportive care, along with early chelation, preferably with succimer or 2,3-dimercapto-1-propane sulfonic acid (DMPS), be given for the patients with this potentially life-threatening poisoning [6].

Pregnant rats injected with 0.5 or 2.5 μmol/kg bw $HgCl_2$ for 6 or 48 hours (h) show rapid accumulation of $Hg^{+2}$, which is dose-dependent, in the placenta. If low dose of $HgCl_2$ is exposed to the mother, accumulation of $Hg^{+2}$ in the fetus increases between 6 hours and 48 hours; meanwhile, at higher doses, the accumulation is the same for each time of observation. The highest $Hg^{+2}$ (nmol/g) concentrations in fetal organs are found in the kidney and then in the liver and brain [7]. Meanwhile the low level of prenatal Hg exposure, through seafood consumption by the mother, was found to be positively correlated with children's language and communication skills at the age of five years [8]. Women of reproductive age and pregnant women should avoid exposure to Hg, even at low levels (4.97μg/L), because of their potentially detrimental effects on fetal development [9]. It is known that maternal exposure to $HgCl_2$ during pregnancy and lactation affects offspring immunity and social behavior [10].

Various efforts to prevent Hg exposure in women of childbearing age and pregnant women have been carried out through experimental animals. Detoxification of Hg and reduction of the teratogenic effect of Hg are two similar measures under different conditions, but so far, these two attempts have not been carried out by the same researchers. A study investigating the protective effect of omega-3 fatty acids (0.5g/kg body weight [bw]/day) on the toxicity of $HgCl_2$ (0.4 mg/kg bw/day) in mice, and malondialdehyde levels, reduced glutathione, oxides. nitric, and total sialic acid have been performed. The results revealed that omega-3 fatty acids could attenuate $HgCl_2$-induced toxicity by increasing antioxidant status and acute-phase response in mice [11]. It was also reported that origanum oil (5 mg/kg) through its antioxidant potential may possess health-promoting properties and could protect cells from oxidative damage induced by $HgCl_2$ (4 mg/kg bw) in rats [12]. The literature survey revealed that in pre-clinical studies, 27 medicinal plants and 27 natural products exhibited significant mitigation of Hg toxicity in experimental animals [13].

The results of the ethnomedicinal plant study showed that the ethanol extract of the leaves of *Etlingera hemisphaerica* (LE3H; http://www.theplantlist.org/tpl1.1/record/kew-243067; 0.39 mg/g bw) in mice has the potential to reduce blood glucose (36,2%) and triglycerides (21.19%) in mice with hyperglycemia and hypertriglycerides [14]. In addition, Hg detoxification studies showed that LE3H (0.39 mg/g bw) had a hepatoprotective effect to reduce $HgCl_2$ (5 mg/kg bw) toxicity in mice [15]. Previous investigations revealed that administration of $HgCl_2$ increased leukocytes and decreased erythrocytes; meanwhile, giving $HgCl_2$ followed by LE3H treatment can protect blood cell counts and control them. $HgCl_2$ administration triggered the presence

of a new 125 kDa protein and caused overexpression of 48 kDa protein; this protein profile could be protected by LE3H treatment as in the control condition. LE3H shows potential protective effects against $HgCl_2$ toxicity in blood of mice. Therefore, dietary supplements of LE3H may be useful for protecting persons who are exposed to $HgCl_2$ [16]. Our previous study indicated that LE3H decreased malformed living fetus as teratogenicity effects of $HgCl_2$ in mice [17]. There is only one report in the literature on the effect of Hg on bone tissue of mammals, showing that prenatal intoxication of experimental animals with Hg has negative effects on fetal development in rats, delaying ossification long bone [18]. Furthermore, morphologically normal living fetuses (MNLF) were collected from the previously studied and examined whether the fetuses experienced skeletal and/or organ developmental disorders.

Based on the above considerations, the aim of this in vivo biological modeling study was to investigate the effectiveness of LE3H against defects in fetal anatomy and endochondral ossification in mice induced by $HgCl_2$ during the post-implantation period.

## 2. Methods

### 2.1. Extract preparation

This study used *E. hemisphaerica* leaf extract taken from the base of the stem. Plant identification was carried out by the staff of the Plant and Botanical Garden Conservation Research Center, Indonesian Institute of Sciences (http://lipi.go.id/). The plant materials were collected from the surroundings of Curup City, Rejang Lebong District, Bengkulu Province, Indonesia. Furthermore, the leaves are washed and cut into small pieces. *E. hemisphaerica* fresh leaves weighing 3000 g were wind-dried for 7 days, resulting in 800 g of dry leaves which were blended into a fine powder. Leaf powder was macerated in 2 L of ethanol (96%) for 7 days, and the resulting filtrate was evaporated using a rotary evaporator [19] so that the concentrated extract was weighed 3 g, and 2 g was used as the test material for this study [15–17].

### 2.2. Dosages of investigation

Based on the results of previous studies [16, 17], this study used a single dose of *E. hemisphaerica* leaf ethanol extract (LE3H), namely 0.39 mg/g bw [15]. Hg in the form of $HgCl_2$ was obtained from Merck (Germany; Product No. 104 417). The dose of $HgCl_2$ used in this study was 5 mg/kg bw [20, 21].

### 2.3. Group of Experimental Animals (GEA)

Swiss Webster mice (*Mus musculus*) from Animal Test Center, School of Life Sciences and Technology (SITH; https://www.itb.ac.id/sekolah-ilmu-dan-teknologi-hayati), Bandung Institute of Technology (ITB) used as an experimental animal.

Rearing of the animals was done in a room at 23–27˚C and 83% humidity. Food and water were given *ad libitum*. When female mice achieved their sexual maturity (10–12 weeks old) they were mated with a male (1:1). A vaginal plug detected in the following morning was defined as 0 gestation day (GD) [22].

Pregnant mice were divided into four groups, each consisting of 10 dams. The first group was given LE3H (E1), the second group was $HgCl_2$ (E2), the third group was $HgCl_2$+LE3H (E3), and the fourth group was control (E0), only double distilled water was given. $HgCl_2$ (5 mg/kg bw) was given by injection at GD9 and LE3H (0.39 mg/g bw) was given gavage on GD10.

## 2.4. Evaluation of maternal, embryonic, and fetal toxicity

Animals were killed by dislocating the cervix at GD18, surgically removed, and the uterus removed and cleaned with saline (0.9% NaCl) solution. Cases of resorption, intra-uterine death, and live fetuses were recorded for each dam. The living fetus (LF) was cleaned with NaCl, weighed, and measured, and the fetal morphological condition (normal/abnormal) was observed and compared with E0 as an indicator of the degree of teratogenicity [22] at E1, E2, and E3. LF was collected from each group of experimental animals and divided into two categories, namely morphological normal living fetus (MNLF) and malformed living fetus (MLF) [17]. MNLF from each of the 5 dams of E0, E1, E2, and E0 were used as material test for soft tissue examination, and then the remaining were used for skeletal examination.

## 2.5. Soft tissue examination

The MNLF from the 5 dams of E0, E1, E2, and E3 were collected for anatomical observation. The *Bouin* solution was diluted firstly with a ratio of 15 (*Bouin* solution): 5 (formalin): 1 (acetic acid) and allowed to stand for 24 hours. After the *Bouin* solution was ready, all the fetal mice from one parent were inserted into the bottle and the entire soft tissues were soaked for 14 days. The fetuses were lifted from the bottle, rinsed twice with water, and then immersed with 70% ethanol before sieving [22]. Transverse incisions through the 8 (eight) points of fetus body with a free hand razor blade were done to observe difference between the treated groups and the control group (Fig 1; [22, 23]).

## 2.6. Skeletal examination

The MNLF from the 5 dams of E0, E1, E, and E3 were fixed with 96% ethanol for one week, and then stained using *Alizarin Red S* and *Alcian Blue* for ossification analysis. Cartilage and mineralized bone were characterized with different colors (blue and red, respectively) after the staining according to standard procedures [24]. Humerus dan femur bones were isolated from stained bone fetuses, and then their indexes of length and width of ossified parts were determined as follows:

**2.6.1. Index of length of ossified part (growth in length, ILOP).** The length of bone (A) was measured between two tips of the bone with a caliper. Meanwhile, ossified part (B) was measured on stained (red) area of the bone using the same caliper. Bone length measurements were made for the left and right, and then the average length value was used. Furthermore, index of length of ossified part was determined by dividing the length with the ossified part (B/A; Figs 2 and 5; [25]).

**2.6.2. Index of width of ossified part (growth in diameter, IWOP).** Cross-section was made at the middle of the bone, and then the obtained ring was measured with a micrometer under a microscope. Bone diameter measurements were done for the left and right, and then the average diameter was calculated. The diameter of cross-section (c) and lumen of the ring (d) were used for calculating the width of the cross-section (C) and the lumen (D) respectively. The width of ossified part (E) was obtained by substracting the width of the cross-section with the lumen (C-D). Finally, the index of width of ossified part was determined by dividing the width of ossified part with the width of cross-section (E/A; Figs 2 and 5; [25]).

## 2.7. Statistical analyses

The data obtained from this study were generalized by the $\chi^2$ test of goodness of fit (Table 1) and by multiple comparation, and then the least significant difference (Tables 3–6) [26].

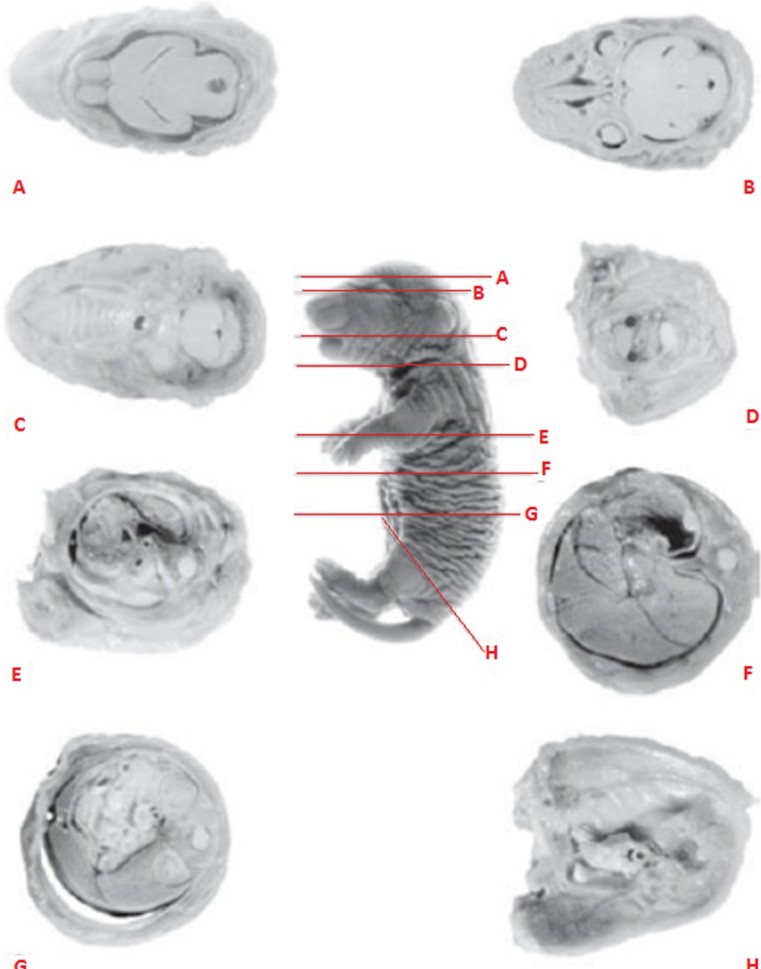

**Fig 1. Transversal sections (A–H) of a 21-day-old male rat fetus (Bouin's fixation, Wilson's free-hand method, with own modification for the head examination [22].**

## 2.8. Ethical statement

This study was carried out in strict accordance with the recommendations in the Guide for the Care and Use of Laboratory Animals of the National Institutes of Health This study was

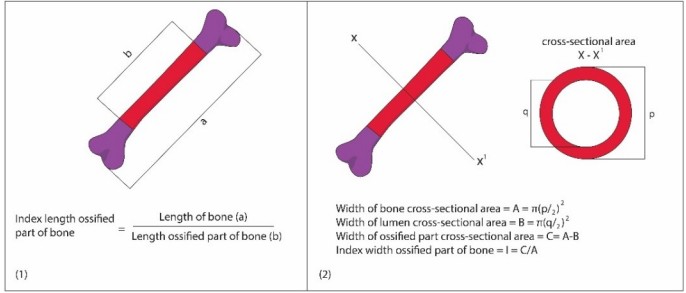

**Fig 2. Determining the Index of Length of Ossified Part (ILOP; 1) and Index of Width Of Ossified Part (IWOP; 2) of humerus and femur in Morphologically Normal Living Fetuses (MNLF) of *M. musculus* on gestation day (GD18; [24]; http://journal.itb.ac.id/index.php?li = article_detail&id = 625).**

**Table 1. Results of soft tissue examination using the free hand razor blade technique through eight points of fetus body in Morphological Normal Live Fetus (MNLF) of *M. musculus* on Gestation Day (GD) 18 which previously were given 0.39 mg/kg bw LE3H through gavage on GD10(E1), injected with 5 mg/kg bw HgCl$_2$ on GD9(E2), and administered 5 mg/g bw HgCl$_2$ on GD9 and then 0.39 mg/kg bw LE3H on GD10(E3).** Meanwhile the controls (E0) were administered double-distilled water only.

| Group of Experimental Animals (GEA) | Number of Dams | Morphologically Normal Living Fetuses (MNLF; %) | Anatomy | |
|---|---|---|---|---|
| | | | Normal (%) | Defective (%) |
| E0: Controls were administered double-distilled water only. | 5 | 48 (100.0) | 48 (100.0) | 0 (0.0) |
| E1: 0.39 mg/g bw LE3H on GD 10 (gavage) | 5 | 40 (100.0) | 34 (85.0) | 6 (15.0) |
| E2: 5 mg/kg bw HgCl$_2$ on GD 9 (ip) | 5 | 35 (100.0) | 9 (25.7) | 26 (74.3) |
| E3: 5 mg/kg bw HgCl$_2$ on GD 9 (ip) + 0.39 mg/g bw LE3H on GD 10 (gavage) | 5 | 45 (100.0) | 23 (51.1) | 22 (48.9) |

**Notes**: $\chi^2$ = 69.24; There are significant differences between normal and defective anatomy E0, E1, E2, and E3 (p < 0.05; [26]).

conducted by following the ethics of animal use, including aspects of the humane treatment of animals, in accordance with the principle of 5F (Freedom), namely; (a) free from hunger and thirst, (b) free from discomfort, (c) free from pain, injury and disease, (d) free from fear and the long-term stress, (e) freely expressing behavior naturally, given space and appropriate facilities [27–29]. The protocol was approved by the Committee on the Ethics of Animal Experiments of Bengkulu University.

## 3. Results

### 3.1. Fetal anatomy

The morphological studies in modern teratological investigations recommend transverse and sagittal incisions for soft tissue examination [22], but because of the inadequacy of the number of fetuses in this study, only transversal incisions were done. Transversal sections trough (A) brain, (B) eye, (C) oral cavity and digestive tract, (D) esophagus and throat, (E) spine, lungs and heart, (F) bile, liver, and stomach, (G) small intestine and colon, and (H) spine and the anal canal were observed from the morphologically normal living fetuses (MNLF) of *M. musculus* on GD 18. In this study we did not find any anatomical defect on incisions F and H, so six types of incisions are presented (Fig 3).

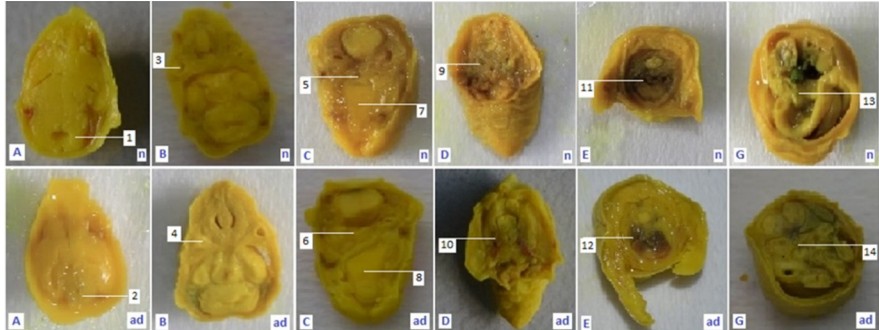

**Fig 3. Results of soft tissue examination (see also Fig 1) using the freehand razor blade technique through eight points of fetus body on gestation day (GD) 18.** Description, sagittal section trough: A: Brain; B: Eye; C: Oral cavity and digestive tract; D: Esophagus and throat; E: Spine, lungs and heart; G: Small intestine and colon; n: normal anatomy; ad: defective anatomy; 1: normal hind brain; 2: small hind brain; 3: normal len vesicle; 4: small len vesicle; 5: normal respiratory tract; 6: narrow respiratory tract; 7: normal palate; 8: wavy palate; 9: normal throat channel; 10: narrow throat channel; 11: normal trachea; 12: narrow trachea; 13: normal small intestine; 14: narrow small intestine.

The six types of incisions were generated and then compared to similar incisions of the control compound so that they could be grouped into the criteria of normal or anatomically defective. The results of non-parametric statistical calculations showed that the treatments had a significant effect ($\chi^2 = 69.24$) on the number of cases of anatomical defects. E2 had anatomical defect cases of 74.6%, higher than that of E3 (48.9%), E1 (15.0%) and E0 (0%). Thus, LE3H has the capacity to reduce the number of anatomical defects due to $HgCl_2$ treatment (Table 1 and Fig 4).

Results of soft tissue examination revealed fifty-four (54) anatomical defects from four groups of experimental animals (GEA) with ninety-eight (98) defect cases, indicating that one individual fetus may have more than one type of anatomical defect. The sequence of frequencies recorded from this study was as follows: (B) eye 24, (A) brain 21, (G) small intestine and colon 21, (C) oral cavity and digestive tract 11, (E) spine, lungs and heart 11, and (D) esophagus and throat 10. E1 produced 8 anatomical defects, E2 57, and E3 33. These observations revealed that LE3H was capable of reducing cases of anatomical defect due to $HgCl_2$ treatment (Table 2).

## 3.2. Endochondral ossification

**3.2.1. The Index of Length of Ossified Part (ILOP) of humerus.**   The index of length of ossified part (growth in length, ILOP) humerus in *M. musculus* fetuses on gestation day (GD) 18 showed that E1 had ILOP of 0.68±0.07, greater than that of E0 (0.58±0.13) and E2 (0.59 ±0.13), which was not significantly different from that of E0. Furthermore, ILOP in E3 (0.63 ±0.23) was significantly higher than that of E0, but not significantly different from that of E1 and E2 (Figs 5 and 6 and Table 3).

**3.2.2. The Index Width of Ossified Part (IWOP) of humerus.**   The index of width of ossified part (growth in diameter, IWOP) humerus in *M. musculus* fetuses on GD 18 showed that E2 had IWOP of 0.82±0.03, which was significantly lower than that E0 (0.89±0.04) and E1 (0.89±0.03), while that of E1 and E0 was not significantly different from one another.

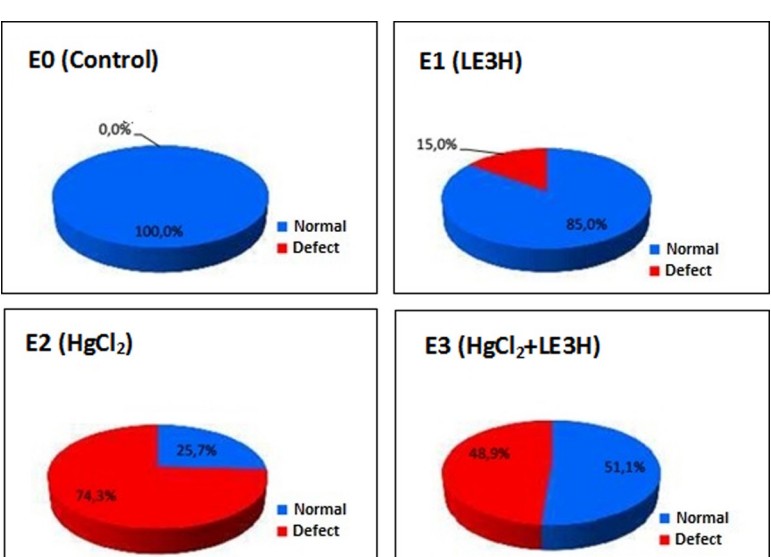

**Fig 4. Percentage normal and defective anatomy in Morphologically Normal Living Fetuses (MNLF) of *M. musculus* on Gestation Day (GD)18 which previously were given 0.39 mg/kg bw LE3H through gavage on GD10 (E1), injected with 5 mg/kg bw HgCl₂ on GD9(E2), and administered 5 mg/kg bw HgCl₂ on GD9 and then 0.39 mg/g bw LE3H on GD10(E3).** Meanwhile the controls (E0) were administered double-distilled water only.

**Table 2. Results of soft tissue examination using the free hand razor blade technique through eight points of fetus body on Gestation Day (GD) which previously were given 0.39 mg/g bw LE3H through gavage on GD10 (E1), injected with 5 mg/kg bw HgCl₂ on GD9 (E2), and administered 5 mg/kg bw HgCl₂ on GD9 and then 0.39 mg/kg bw LE3H on GD10 (E3).** Meanwhile the controls (E0) were administered double-distilled water only.

| Group of Experimental Animals (GEA) | Number of Dam | Anatomical Defect (AD; %) | Kind of anatomical defect | | | | | | | | |
|---|---|---|---|---|---|---|---|---|---|---|---|
| | | | Sum of AD | A | B | C | D | E | F | G | H |
| E0: Controls were administered double-distilled water only. | 5 | 0 (0.0) | 0 | 0 | 0 | 0 | 0 | 0 | 0 | 0 | 0 |
| E1: 0.39 mg/g bw LE3H on GD 10 (gavage) | 5 | 6 (15.0) | 8 | 0 | 5 | 0 | 0 | 0 | 0 | 3 | 0 |
| E2: 5 mg/kg bw HgCl₂ on GD 9 (ip) | 5 | 26 (74.3) | 57 | 15 | 10 | 7 | 7 | 6 | 0 | 12 | 0 |
| E3: 5 mg/kg bw HgCl₂ on GD 9 (ip) + 0.39 mg/g bw LE3H on GD 10 (gavage) | 5 | 22 (48.9) | 33 | 6 | 9 | 4 | 3 | 5 | 0 | 6 | 0 |

**Notes**: Fre: Frekuensi. Sagittal section through A: Brain; B: Eye; C: Oral cavity and digestive tract; D: Esophagus and throat; E: Spine, lungs and heart; F: Bile, liver, and stomach; G: Small intestine and colon; H: Spine and the anal canal.

Meanwhile, IWOP in E3 (0.88±0.03) was significantly higher than that in E2, but not different from that in E1 and E0 (Table 4 and Fig 7). These facts revealed that HgCl₂ decreased IWOP humerus, but the LE3H could mitigate the effect of HgCl₂.

**3.2.3. The Index of Length of Ossified Part (ILOP) of femur.** The ILOP (growth in length) of femur in *M. musculus* fetuses on GD 18 treated with HgCl₂(E2) was 0.52±0.17 and LE3H(E1) was 0.57±0.10, and both were significantly lower than that of the control (0.64 ±0.09). ILOP in E3(HgCl₂+LE3H) was 0.53±0.15, significantly lower than that of E0 and E1, but not significantly different from that in E2 (Table 5 and Fig 8). These facts revealed that

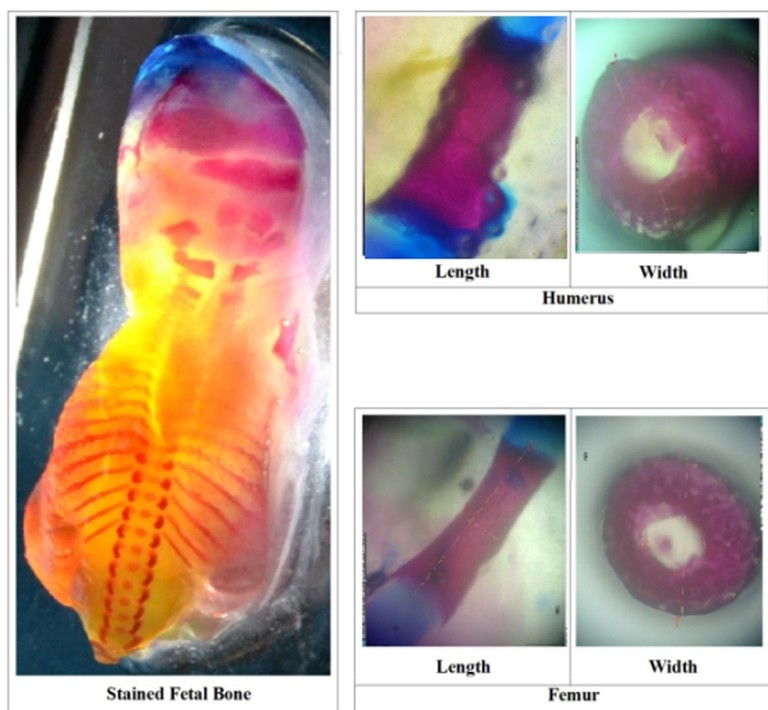

**Fig 5. A sample of stained fetal bone with *Alizarin Red S* and *Alcian Blue* (the size of these images is disproportionate).** The dorsal stained fetal bones of four groups of experimental animals were observed to determine the number of ossified skulls, vertebrae, and ribs on gestation day (GD)18. The index of length of ossified part (ILOP, growth in length) and index of width of ossified part (IWOP, growth in diameter) of humerus and femur in morphologically normal living fetuses (MNLF) of *M. musculus* on GD18 were determined according to Fig 2.

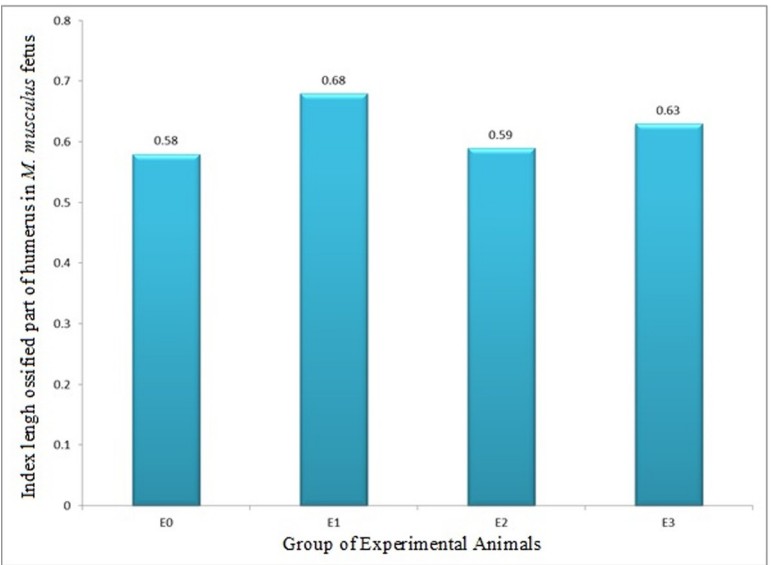

**Fig 6. Index of length of ossified part (growth in length, ILOP) of humerus in *M. musculus* fetuses on Gestation Day (GD) 18 which previously were given 0.39 mg/g bw LE3H through gavage on GD 10 (E1), injected with 5 mg/kg bw HgCl$_2$ on GD 9 (E2), and administered 5 mg/kg bw HgCl$_2$ on GD 9 and then 0.39 mg/g bw LE3H on GD 10 (E3).** Meanwhile, the controls (E0) were administered double-distilled water only.

LE3H alone and HgCl$_2$ alone significantly decreased ILOP of femur, but the combination of HgCl$_2$+LE3H, produced higher ILOP than HgCl$_2$ alone.

**3.2.4. The Index of Width of Ossified Part (IWOP) of femur.** The IWOP (growth in diameter) of femur in *M. musculus* fetuses on GD 18 showed in all groups of experimental animals (GEA) did not statistically show any significant difference (Table 6 and Fig 9) from one another. The facts indicated that LE3H, HgCl$_2$, and HgCl$_2$+LE3H had no significant effect on IWOP of femur.

## 4. Discussion

It should be noted that this teratological research consists of four groups of experimental animals (GEA), namely the first group was administered LE3H (E1), the second one HgCl$_2$ (E2), the third one HgCl$_2$+LE3H (E3), and the fourth was control (E0), administered double-

**Table 3. Length of humerus bone of *M. musculus* fetuses on Gestation Day (GD) 18 which previously were given 0.39 mg/g bw LE3H through gavage on GD 10 (E1), injected with 5 mg/kg bw HgCl$_2$ on GD 9 (E2), and administered 5 mg/kg bw HgCl$_2$ on GD 9 and then 0.39 mg/g bw LE3H on GD 10 (E3).** Meanwhile, the controls (E0) were administered double-distilled water only.

| Group of Experimental Animals (GEA) | Number of Dams (N) | Stained Morphologically Normal Living Fetuses (MNLF) | Length (X±SD) | | |
|---|---|---|---|---|---|
| | | | Bone mm [A] | Ossified part mm[B] | Growth in length, ILOP [B/A] |
| E0: Controls were administered double-distilled water only | 5 | 31 | 0.52±0.02 | 0.36±0.06 | 0.58±0.13[a] |
| E1: 0.39 mg/g bw LE3H on GD 10 (gavage) | 5 | 33 | 0.52±0.02 | 0.36±0.03 | 0.68±0.07[c] |
| E2: 5 mg/kg bw HgCl$_2$ on GD 9 (ip) | 5 | 23 | 0.52±0.02 | 0.30±0.06 | 0.59±0.13[ab] |
| E3: 5 mg/kg bw HgCl$_2$ on GD 9 (ip) + 0.39 mg/g bw LE3H on GD 10 (gavage) | 5 | 26 | 0.51±0.06 | 0.31±0.05 | 0.63±0.23[bc] |

Note: Numbers in the same column followed by the same superscript letters ([a, b]) are not significantly different (p < 0.05; [26]). ILOP: Index of length of ossified part.

**Table 4. Width of humerus bone of *M. musculus* fetuses on Gestation Day (GD) 18 which previously were given 0.39 mg/g bw LE3H through gavage on GD 10 (E1), injected with 5 mg/kg bw HgCl$_2$ on GD 9 (E2), and administered 5 mg/kg bw HgCl$_2$ on GD 9 and then 0.39 mg/g bw LE3H on GD 10 (E3).** Meanwhile, the controls (E0) were administered double-distilled water only.

| Group of experimental Animals (GEA) | Number of Dams (N) | Stained Morphologically Normal Living Fetuses (MNLF | Width (X±SD) | | |
|---|---|---|---|---|---|
| | | | Bone mm$^2$[A] | Ossified part mm$^2$[B] | Growth in diameter, IWOP [B/A] |
| E0: Controls were administered double-distilled water only | 5 | 26 | 0.012 ±0.005 | 0.011±0.005 | 0.89±0.04[bc] |
| E1: 0.39 mg/g bw LE3H on GD 10 (gavage) | 5 | 27 | 0.012 ±0.005 | 0.011±0.004 | 0.89±0.03[c] |
| E2: 5 mg/kg bw HgCl$_2$ on GD 9 (ip) | 5 | 18 | 0.010 ±0.006 | 0.010±0.006 | 0.82±0.03[a] |
| E3: 5 mg/kg bw HgCl$_2$ on GD 9 (ip) + 0.39 mg/g bw LE3H on GD 10 (gavage) | 5 | 22 | 0.011 ±0.003 | 0.010±0.003 | 0.88±0.03[b] |

Note: Numbers in the same column followed by the same superscript letters ([a, b]) are not significantly different (p < 0.05; [26]); Index of width of ossified part.

distilled water only. Each GEA consists of 10 dams. The living fetus (LF) was collected from each GEA, and divided into two categories; malformed living fetus (MLF) and morphological normal living fetus (MNLF) [17]. MLF data has been separately reported and can be accessed via the following webs (https://pubmed.ncbi.nlm.nih.gov/32257925/; and/or DOI: 10.1007/s43188-019-00010-8). This paper will focus on discussing data about the soft tissue examination and skeletal examination of MNLF.

This study did not find anatomical defects on incisions F (bile, liver, and stomach) and H (spine and the anal canal), therefore here six types of incisions were presented (Fig 3). Based on the six types of incisions it was found that HgCl$_2$ resulted in anatomical defect cases of 74.6%, greater than LE3H (15.0%) and the control (0%). The combination of HgCl$_2$ and LE3H resulted in 48.9% cases of anatomical defects. Thus, LE3H has the capacity to reduce the

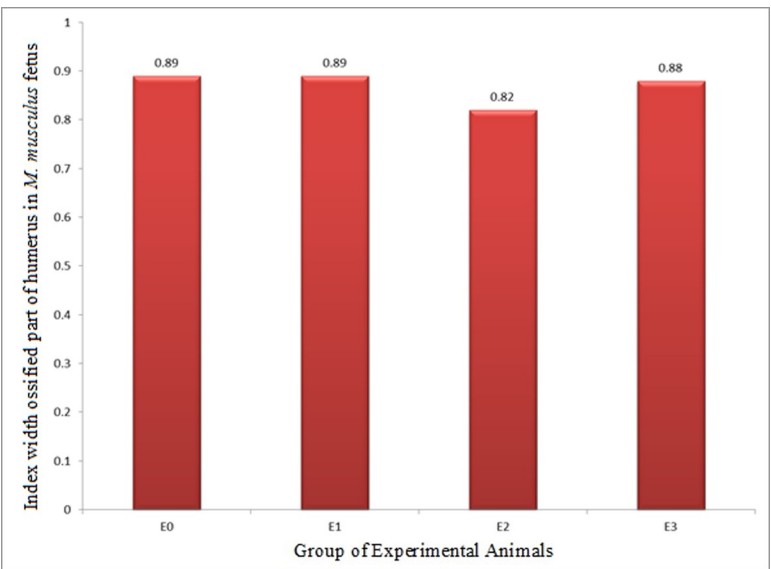

**Fig 7. Index of width of ossified part (growth in diameter, IWOP) of humerus in *M. musculus* fetuses on gestation day (GD) 18 which previously were given 0.39 mg/g bw LE3H through gavage on GD 10 (E1), injected with 5 mg/kg bw HgCl$_2$ on GD 9 (E2), and administered 5 mg/kg bw HgCl$_2$ on GD 9 and then 0.39 mg/g bw LE3H on GD 10 (E3).** Meanwhile, the controls (E0) were administered double-distilled water only.

**Table 5. Length of femur bone of *M. musculus* fetuses on Gestation Day (GD) 18 which previously were given 0.39 mg/g bw LE3H through gavage on GD 10 (E1), injected with 5 mg/kg bw HgCl₂ on GD 9 (E2), and administered 5 mg/kgbw HgCl₂ on GD 9 and then 0.39 mg/g bw LE3H on GD 10 (E3).** Meanwhile, the controls (E0) were administered double-distilled water only.

| Group of experimental animals (GEA) | Number of Dams (N) | Stained Morphologically Normal Living Fetuses (MNLF) | Length (X±SD) | | |
|---|---|---|---|---|---|
| | | | Bone mm [A] | Ossified part mm[B] | Growth in length, ILOP [B/A] |
| E0: Controls were administered double-distilled water only | 5 | 31 | 0.52±0.02 | 0.31±0.05 | 0.64±0.09[d] |
| E1: 0.39 mg/g bw LE3H on GD 10 (gavage) | 5 | 33 | 0.52±0.03 | 0.30±0.04 | 0.57±0.10[bc] |
| E2: 5 mg/kg bw HgCl₂ on GD 9 (ip) | 5 | 23 | 0.52±0.01 | 0.26±0.08 | 0.52±0.17[a] |
| E3: 5 mg/kg bw HgCl₂ on GD 9 (ip) + 0.39 mg/g bw LE3H on GD 10 (gavage) | 5 | 26 | 0.51±0.02 | 0.27±0.07 | 0.53±0.15[ab] |

Note: Numbers in the same column followed by the same superscript letters ([a, b]) are not significantly different (p < 0.05; [26]); ILOP: Index of length of ossified part.

number of anatomical defects due to HgCl₂ treatment up to 24.8% (Table 1 and Fig 4). Results of the soft tissue examination revealed that one individual fetus may bear more than one type of anatomical defect. HgCl₂ resulted in 57 cases of anatomical defects, while the combination of LE3H and HgCl₂ produced 33 cases. So, LE3H was capable of reducing 24 cases of anatomical defect (42,1%) due to HgCl₂ (Table 2). The treatment of LE3H alone resulted in 8 cases of anatomical defects, higher than that in the control (0%). In the HgCl₂+LE3H group of experimental animals, ip treatment was 5 mg/kg bw HgCl₂ on GD 9 followed by gavage administration of 0.39 mg/g bw LE3H on GD 10, so there was an interval of less than 48 h. Previous similar trials reported that when dams were exposed to the dose of HgCl₂, fetal accumulation of Hg²⁺ increased between 6–48 h, within fetal organs; the greatest concentration of Hg²⁺ was found in the kidneys, followed by the liver and brain. A dose-dependent increase in the accumulation of Hg²⁺ in fetal organs was observed, suggesting that continued maternal exposure may lead to increased fetal exposure. These data indicate that Hg²⁺ is capable of crossing the

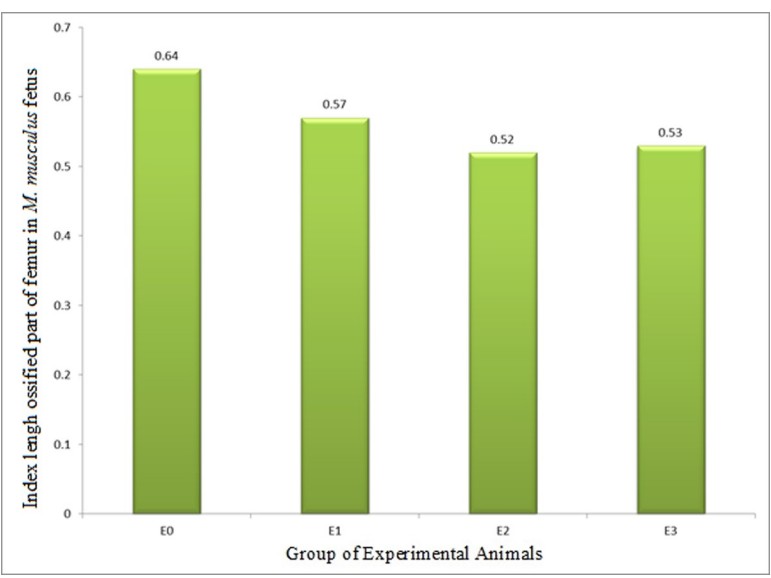

**Fig 8. Index of length of ossified part (growth in length, ILOP) of femur in *M. musculus* fetuses on gestation day (GD) 18 which previously were given 0.39 mg/g bw LE3H through gavage on GD 10 (E1), injected with 5 mg/kg bw HgCl₂ on GD 9 (E2), and administered 5 mg/kg bw HgCl₂ on GD 9 and then 0.39 mg/g bw LE3H on GD 10 (E3).** Meanwhile, the controls (E0) were administered double-distilled water only.

**Table 6. Width of femur bone of *M. musculus* fetuses on Gestation Day (GD) 18 which previously were given 0.39 mg/g bw LE3H through gavage on GD 10 (E1), injected with 5 mg/kg bw HgCl₂ on GD 9 (E2), and administered 5 mg/kg bw HgCl₂ on GD 9 and then 0.39 mg/g bw LE3H on GD 10 (E3).** Meanwhile, the controls (E0) were administered double-distilled water only.

| Group of experimental animals (GEA) | Number of Dams (N) | Stained Morphologically Normal Living Fetuses (MNLF) | Width (X±SD) | | |
|---|---|---|---|---|---|
| | | | Bone mm$^2$[A] | Ossified part mm$^2$[B] | growth in diameter, IWOP[B/A] |
| E0: Controls were administered double-distilled water only | 5 | 26 | 0.46±0.007 | 0.016±0.007 | 0.92±0.35[a] |
| E1: 0.39 mg/g bw LE3H on GD 10 (gavage) | 5 | 28 | 0.42±0.006 | 0.014±0.006 | 0.91±0.36[a] |
| E2: 5 mg/kg bw HgCl₂ on GD 9 (ip) | 5 | 19 | 0.29±0.006 | 0.014±0.006 | 0.91±0.32[a] |
| E3: 5 mg/kg bw HgCl₂ on GD 9 (ip) + 0.39 mg/g bw LE3H on GD 10 (gavage) | 5 | 21 | 0.31±0.006 | 0.013±0.006 | 0.90±0.35[a] |

Note: Numbers in the same column followed by the same superscript letters ([a, b]) are not significantly different (p < 0.05; [26]); IWOP: Index width ossified part.

placenta and gaining access to fetal organs in a dose-dependent manner [7]. It has been previously reported that serum activities of alkaline phosphatase, alanine aminotransferase, aspartate aminotransferase, glutamyl-transferase, and lactate dehydrogenase were significantly higher, while serum levels of total protein, albumin, triglyceride, total cholesterol, and low-density lipoprotein cholesterol were significantly lower in the HgCl₂-treated animals than in the control. Malondialdehyde level significantly increased and superoxide dismutase, catalase, and glutathione peroxidase activities decreased in liver tissue of HgCl₂-treated mammal. HgCl₂ exposure in histopathological changes [30]. The histopathological changes obtained in this study are: a. small hind brain; b. small lens vesicle; c. narrow respiratory tract; d. wavy palate; e. narrow throat channel; f narrow trachea; g. narrow small intestine (Fig 3 and Table 2). Supplementation of LE3H provided partial protection in biochemical parameters that were altered by HgCl₂. As a result, LE3H significantly reduced HgCl₂-induced fetal anatomy toxicity.

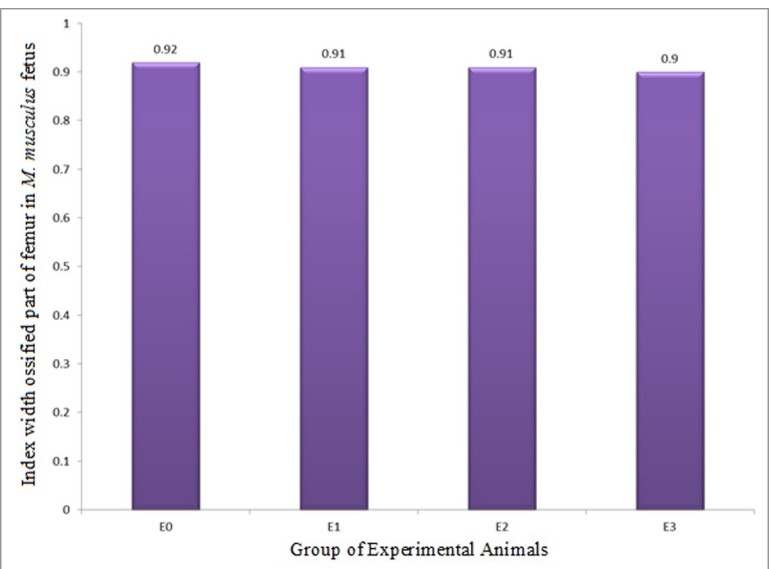

**Fig 9. Index of width of ossified part (growth in diameter, IWOP) of femur in *M. musculus* fetuses on gestation day (GD) 18 which previously were given 0.39 mg/g bw LE3H through gavage on GD 10 (E1), injected with 5 mg/ kg bw HgCl₂ on GD 9 (E2), and administered 5 mg/kg bw HgCl₂ on GD 9 and then 0.39 mg/g bw LE3H on GD 10 (E3).** Meanwhile, the controls (E0) were administered double-distilled water only.

The endochondral ossification is an indirect ossification process. Initially, mesenchyme gives rise to a temporary hyaline cartilage precursor. Cartilage is eventually replaced by bone in the epiphysis and diaphysis, apart from the epiphyseal plate region [31]. Here the bone continues to grow in length until maturity. The periosteum comes from the perichondrium which is a membrane made of connective tissue that covers cartilage during endochondral ossification at the bone collar. The periosteum contains a layer of undifferentiated cells (then the collection of cells is called as osteoprogenitor cells) that allow the growth in bone diameter (https://www.emouseatlas.org/emap/home.html). During the post-implantation period in mice, there were four parameters in this endochondral ossification research, namely: (1) growth in length of humerus (ILOP humerus); (2) growth in diameter of the humerus (IWOP humerus); (3) growth in length of femur (ILOP femur); and (4) growth in diameter of femur (IWOP femur). Treatment by $HgCl_2$ produced different effects on the humerus and femur. The treatment of $HgCl_2$ did not result in different growth in length of humerus (0.59±0.13) compared to the control (0.58±0.13; Table 3 and Fig 6), but significantly decreased growth in diameter of humerus by 7.8%, i.e., from 0.89±0.04 in the control to 0.82±0.03 in E1 (Table 4 and Fig 7). The femur given the treatment of $HgCl_2$ on GD 9 had significantly lower growth in length of femur 0.52±0.17 than the control 0.64±0.09, or 18.7% decrease (Table 5 and Fig 8), but had different growth in diameter (0.91±0.32) compared to controls (0.92±0.35; Table 6 and Fig 9). We suspect that the effects were different because the humerus (forelimb) develops earlier than the femur (hindlimb) [32], and it has also been known that growth in length develops much earlier than growth in diameter in the long bones [33]. Different stages of development lead to different sensitivity to external stimuli. One of the four parameters of endochondral ossification, namely the IWOP humerus, shows the antagonistic phenomenon between $HgCl_2$ and LE3H during the post-implantation period in mice. These facts revealed that LE3H did not result in higher IWOP humerus than that in the control, while $HgCl_2$ significantly resulted in lower IWOP humerus than that in the control, whereas $HgCl_2$+LE3H did not produce significant difference in IWOP humerus from the control and LE3H (Table 4 and Fig 7). Previous studies reported that $HgCl_2$ led to cell death, reactive oxygen species (ROS) increase, and cytosolic caspase-3 activation. The ROS increase was related to the decreased level of glutathione (GSH). Serum lactate dehydrogenase (LDH) and tumor necrosis factor alpha (TNF-α) were higher in the Hg group than in the control group. Chromatin condensation evaluated by 4,6-diamidino-2-phenylindole (DAPI) staining were also detected in mercury-treated cells and this suggest the apoptotic process of cells by $HgCl_2$ [34]. Changes in oxidant-antioxidant balance, inhibition of antioxidant defense system, and enhanced production of ROS are considered to play a key role in Hg-induced toxicity [35]. LE3H is a crude extract containing flavonoids [16], and that is thought to include flavonoid quercetin. Several studies have reported that the flavonoid quercetin has antioxidant effects against $HgCl_2$ [36–39]. As a result, LE3H significantly reduced $HgCl_2$-induced defects in the IWOP humerus.

The flavonoid quercetin is part of results from a two-decade literature survey (1998–2018) which revealed that in pre-clinical studies, 27 medicinal plants and 27 natural products exhibited significant mitigation from Hg toxicity in experimental animals. The literature survey of those experimental studies on medicinal plants and phytochemicals with ameliorative effects on mercury toxicity produces more discussion about antioxidant activity than mercury chelating activity. Of the ninety of the literatures reviewed [13] only three (3.3%) discuss mercury chelating activity [40–42]. Twenty-seven (27) plant-derived natural products were found to alleviative effects of mercury, and four of these natural materials need to be recommended for further comprehensive clinical exploitation [13], namely ascorbic acid [43–45], α tocopherol [46], beta carotene [47], and the flavonoid quercetin [48]. The flavonoid quercetin is a typical flavonoid, possesses diverse biochemical and physiological actions, including antiplatelet,

estrogenic, and anti-inflammatory properties [49]. Two researches indicated the reduction in toxicity and teratogenicity of $HgCl_2$ by administering LE3H to the animals [16, 17]. LE3H is additional information for ameliorative effects on mercury toxicity. Meanwhile the presence of ascorbic acid, α tocopherol, and beta carotene on LE3H should be further detected.

### 4.1. Study limitations

The results of this study indicated that in MNLF there were defects in fetal anatomy and endochondral ossification due to the treatment of $HgCl_2$, and the condition was likely to be recovered by the provision of natural material, LE3H. The fact shows that LE3H is a prospective natural ingredient to repair the damaging effects of Hg pollution in the environment. Meanwhile, the data collected from in vivo biological modeling studies have not been able to explain the molecular mechanisms in reducing the teratogenic effects of $HgCl_2$ by administering LE3H. The further molecular mechanism still needs to be studied in more detail at the protein level; it may be performed through a teratoproteomics analysis [28, 50].

## 5. Conclusion

Leaf ethanolic extract of *E. hemisphaerica* (LE3H; 0.39 mg/g bw) is a prospective natural product which could significantly mitigate defects of fetal anatomy and endochondral ossification induced by mercuric chloride ($HgCl_2$; 5 mg/kg bw) on *Mus musculus* during post-implantation period.

## Acknowledgments

We express our gratitude to Dr. Choirul Muslim, for his valuable recommendations for conducting the research. We also gratefully acknowledge Nurus Sa'adiyah for her statistical analysis, Shafira R. Ruyani for her illustration, Dr. Wiryono (Bengkulu University, Indonesia) and Dr. Corey Johnson (University of North Carolina at Greensboro, USA) for their grammatical suggestions in preparing this manuscript.

## Author Contributions

**Conceptualization:** Aceng Ruyani, Agus Sundaryono, Agus Susanta.

**Data curation:** Eda Kartika, Deni Parlindungan, Riza Julian Putra.

**Formal analysis:** Aceng Ruyani, Agus Susanta.

**Funding acquisition:** Aceng Ruyani.

**Investigation:** Eda Kartika, Deni Parlindungan, Riza Julian Putra.

**Methodology:** Deni Parlindungan, Riza Julian Putra.

**Project administration:** Eda Kartika, Deni Parlindungan, Riza Julian Putra.

**Resources:** Aceng Ruyani, Agus Sundaryono, Agus Susanta.

**Supervision:** Agus Sundaryono, Agus Susanta.

**Validation:** Aceng Ruyani.

**Writing – original draft:** Aceng Ruyani.

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
