## [Decision Letter · Decision Letter 0]

20 Oct 2020

PONE-D-20-28046

Leaf ethanolic extract of Etlingera hemesphaerica Blume mitigates fetal anatomy and endochondral ossification-induced mercuric chloride during the post-implantation period in Mus musculus

PLOS ONE

Dear Dr. Ruyani,

Thank you for submitting your manuscript to PLOS ONE. After careful consideration, we feel that it has merit but does not fully meet PLOS ONE’s publication criteria as it currently stands. Therefore, we invite you to submit a revised version of the manuscript that addresses the points raised during the review process.

The reviewers recognized that your study provies technically and scientifically useful contributions. However, there were significant concerns noted. The English language will require extensive editing to make the manuscript clear and more readable. Clarifying statistical methods is needed, and you should clarify the confusion that the leaf extract induced bone defects. Finally, additional experiments were requested to visualize tissue status and measure bone cell activities.  I support the reviewers' assessment and feel that these revisions are required.

We look forward to receiving your revised manuscript.

Kind regards,

James A. Marrs

Academic Editor

PLOS ONE

Journal Requirements:

"Swiss Webster mice (Mus musculus) from the Animal Test Center, the School of Life Sciences and Technology (SITH; https://www.itb.ac.id/sekolah-ilmu-dan-teknologi-hayati), Bandung Institute of Technology (ITB) were used as experimental animals were recommended by local regulations established by the Institutional Animal Care and Use Committee (IACUC) of Bengkulu University (Komisi Bioetika, Universitas Bengkulu). This study paid attention to the ethical use of animals including aspects of the humane treatment of animals, in accordance with the principle of 5F (Freedom), namely; (a) free from hunger and thirst, (b) free from discomfort, (c) free from pain, injury and disease, (d) free from fear and the long-term stress, (e) freely express behavior naturally, given space and appropriate facilities [22; 23;24].".   

i) Please amend your current ethics statement to confirm that your named ethics committee specifically approved this study.

For additional information about PLOS ONE submissions requirements for ethics oversight of animal work, please refer to http://journals.plos.org/plosone/s/submission-guidelines#loc-animal-research  

ii) Once you have amended this/these statement(s) in the Methods section of the manuscript, please add the same text to the “Ethics Statement” field of the submission form (via “Edit Submission”).

3. Please include your tables as part of your main manuscript and remove the individual files. Please note that supplementary tables (should remain/ be uploaded) as separate "supporting information" files

5. Thank you for submitting the above manuscript to PLOS ONE. During our internal evaluation of the manuscript, we found significant text overlap between your submission and the following previously published works, one of which you are an author:

https://link.springer.com/article/10.1007/s43188-019-00010-8

https://www.tandfonline.com/doi/abs/10.1080/19390211.2018.1429516?journalCode=ijds20

We would like to make you aware that copying extracts from previous publications, especially outside the methods section, word-for-word is    unacceptable. In addition, the reproduction of text from published reports has implications for the copyright that may apply to the publications. Please revise the manuscript to rephrase the duplicated text, cite your sources, and provide details as to how the current manuscript advances on previous work. Please note that further consideration is dependent on the submission of a manuscript that addresses these concerns about the overlap in text with published work. We will carefully review your manuscript upon resubmission, so please ensure that your revision is thorough.

Additional Editor Comments (if provided):

The reviewers strongly recommend English language editing. I support this recommendation. The manuscript must be revised for clarity and grammatical structure.

One reviewer recommends clarifying statistical analysis methods, and the reviewer recommends performing additional experiments to measure bone cellular activities and histological analysis to evaluate tissue status. I also support these recommendations.

Reviewers' comments:

Reviewer's Responses to Questions

**Comments to the Author**

1. Is the manuscript technically sound, and do the data support the conclusions?

Reviewer #1: Partly

Reviewer #2: Yes

2. Has the statistical analysis been performed appropriately and rigorously? 

Reviewer #1: No

Reviewer #2: Yes

3. Have the authors made all data underlying the findings in their manuscript fully available?

Reviewer #1: Yes

Reviewer #2: No

4. Is the manuscript presented in an intelligible fashion and written in standard English?

Reviewer #1: No

Reviewer #2: No

5. Review Comments to the Author

Reviewer #1: This manuscript aimed to examine the protective effect of leaf ethanolic extract on fetal anatomy and endochondral ossification induced by mercury. The authors analyzed skeletal defects based on measurements of bone morphology and geometry. The skeletal defects were not clearly defined. the leaf ethanolic extract (LE3H) is supposed to protect bone development. Instead, LE3H caused several defects. This is confusing. In addition, the statistical analysis was not described. The authors should further examine and compare the cellular activities in bone tissues among the four groups. Histological examination at the cellular level might help to understand the detrimental effects of mercury on skeletal development. The wording in the title should be "induced by mercuric chloride" instead of "induced mercuric chloride"

Reviewer #2: Is the manuscript technically sound, and do the data support the conclusions? (Answer options: Yes, No, Partly)

Yes, The study is well planned and presented in an organized manner, but unfortunately due to language issues, the sentences do not covey the right sense.

Mild statistical analysis has been done and I do not think rigorous analysis is needed.

Have the authors made all data underlying the findings in their manuscript fully available ?

The article in its present form is not suitable for publication AT ALL. The language certainly needs considerable improvement. The sentences need to be phrased according to the established principles of English grammar.

6. PLOS authors have the option to publish the peer review history of their article (what does this mean?). If published, this will include your full peer review and any attached files.

Reviewer #1: No

Reviewer #2: No

---

## [Author Response · Author response to Decision Letter 0]

3 Jan 2021

No. Editor and Reviewer's note Our response

1 We will feel proud if we succeed in publicizing the Open Access PLOS ONE, and understand the provisions of the “Publication fees” that have been set (https://plos.org/publish/fees/).

However, in Bengkulu, Indonesia is one of the people affected by the economic impact of Coved-19, we will ask for “Publication fees” to be lighter.

2 We recommend that you deposit your laboratory protocols in protocols.io to enhance the reproducibility of your results Our laboratory protocols are a normal way of working and do not belong to contemporary work techniques. No need to deposit on protocols.io. What is new in this study is the use of local natural ingredients for the toxicity and teratogenicity of Hg. All laboratory protocols are clearly presented in METHODS from our manuscript.

3 Please ensure that your manuscript meets PLOS ONE's style requirements, including those for file naming. The PLOS ONE style templates can be found. We try to present manuscripts according to PLOS ONE's, and also learn from examples of publications in the last PLOS ONE 

4 Please amend your current ethics statement to confirm that your named ethics committee specifically approved this study 8. Ethical statement 

This study was carried out in strict accordance with the recommendations in the Guide for the Care and Use of Laboratory Animals of the National Institutes of Health This study was conducted by following the ethics of animal use, including aspects of the humane treatment of animals, in accordance with the principle of 5F (Freedom), namely; (a) free from hunger and thirst, (b) free from discomfort, (c) free from pain, injury and disease, (d) free from fear and the long-term stress, (e) freely expressing behavior naturally, given space and appropriate facilities [22;23;24]. The protocol was approved by the Committee on the Ethics of Animal Experiments of Bengkulu University.

5 Please include your tables as part of your main manuscript and remove the individual files. Please note that supplementary tables (should remain/ be uploaded) as separate "supporting information" files This study resulted in eight (6) tables consisting of 2 tables for fetal anatomy, and 4 tables for endochondral ossification. All tables are presented in the results of the manuscript, and all tables are subject to study in the discussion.

6 In your Data Availability statement, you have not specified where the minimal data set underlying the results described in your manuscript can be found. PLOS defines a study's minimal data set as the underlying data used to reach the conclusions drawn in the manuscript and any additional data required to replicate the reported study findings in their entirety. All PLOS journals require that the minimal data set be made fully available. Statement for fetal anatomy: 

“As a result, LE3H significantly reduced HgCl2-induced fetal anatomy toxicity (Table 2), but did not prevent it completely”

Statement for endochondral ossification:

“As a result, LE3H significantly reduced HgCl2-induced defects in the IWOP humerus (Table 4), but did not prevent them completely”.

7 Thank you for submitting the above manuscript to PLOS ONE. During our internal evaluation of the manuscript, we found significant text overlap between your submission and the following previously published works, one of which you are an author. We would like to make you aware that copying extracts from previous publications, especially outside the methods section, word-for-word is unacceptable. In addition, the reproduction of text from published reports has implications for the copyright that may apply to the publications. Please revise the manuscript to rephrase the duplicated text, cite your sources, and provide details as to how the current manuscript advances on previous work. Please note that further consideration is dependent on the submission of a manuscript that addresses these concerns about the overlap in text with published work. We will carefully review your manuscript upon resubmission, so please ensure that your revision is thorough. This manuscript is a follow-up to our previous publication (Ruyani et al., 2019; Ruyani et al., 2020;) entitled "Protective Effect of Leaf Ethanolic Extract Etlingera hemisphaerica Blume Against Mercuric Chloride Toxicity in Blood of Mice" (https://pubmed.ncbi.nlm.nih.gov/29451842/) and "Leaf ethanolic extract of Etlingera hemesphaerica Blume alters mercuric chloride teratogenicity during the post-implantation period in Mus musculus ”(https://pubmed.ncbi.nlm.nih.gov/32257925/).

Living fetuses (LF) were collected from each group of experimental animals and divided into two categories, namely morphologically normal living fetuses (MNLF) and malformed living fetuses (MLF) [Ruyani et al., 2020]. MNLF from each of the 5 dams of E0, E1, E2, and E0 were used as material test for soft tissue examination (fetal anatomy), and then the remaining were used for skeletal examination (endochondral ossification).

Since this manuscript is a follow-up to our previous publication, there must be some similarities in principle. However, similarity language expression is controlled to keep it low using iThenticate software (https://www.ithenticate.com/)

8 The reviewers strongly recommend English language editing. I support this recommendation. The manuscript must be revised for clarity and grammatical structure. Grammatical structure of the manuscript has been double-checked using the Grammarly Insights software (https://app.grammarly.com/)

9 One reviewer recommends clarifying statistical analysis methods, and the reviewer recommends performing additional experiments to measure bone cellular activities and histological analysis to evaluate tissue status. I also support these recommendations 7. Statistical analyses 

The obtained data from this study were generalized by nonparametric and parametric analyses [29].

10 The manuscript must describe a technically sound piece of scientific research with data that supports the conclusions. Experiments must have been conducted rigorously, with appropriate controls, replication, and sample sizes. The conclusions must be drawn appropriately based on the data presented Leaf ethanolic extract of E. hemisphaerica (LE3H; 0.39 mg/g bw) is a prospective natural product which could significantly mitigate defects of fetal anatomy (Table 2) and endochondral ossification (Table 4) induced by mercuric chloride (HgCl2; 5 mg/kg bw) on Mus musculus during post-implantation period but not prevent them completely

11 Have the authors made all data underlying the findings in their manuscript fully available?

 This study resulted in eight (6) tables consisting of 2 tables for fetal anatomy (Table 1; Table 2), and 4 tables for endochondral ossification (Table 3; Table 4, Table 5; Table 6). All tables are presented in the results of the manuscript, and all tables are subject to study in the discussion.

12 Is the manuscript presented in an intelligible fashion and written in standard English? PLOS ONE does not copyedit accepted manuscripts, so the language in submitted articles must be clear, correct, and unambiguous. Any typographical or grammatical errors should be corrected at revision, so please note any specific errors here. Grammatical structure of the manuscript has been double-checked using the Grammarly Insights software (https://app.grammarly.com/)

13 This manuscript aimed to examine the protective effect of leaf ethanolic extract on fetal anatomy and endochondral ossification induced by mercury. The authors analyzed skeletal defects based on measurements of bone morphology and geometry. The skeletal defects were not clearly defined. The leaf ethanolic extract (LE3H) is supposed to protect bone development. Instead, LE3H caused several defects. This is confusing. In addition, the statistical analysis was not described. The authors should further examine and compare the cellular activities in bone tissues among the four groups. Histological examination at the cellular level might help to understand the detrimental effects of mercury on skeletal development. The wording in the title should be “induced by mercuric chloride” instead of “induced mercuric chloride” It should also be noted that the treatment of LE3H alone resulted defects of fetal anatomy and endochondral ossification. So, it can be stated that LE3H has a low teratogenic property in mice. The facts indicate that LE3H is the prospective natural material for ameliorating the detrimental effects of Hg pollution in the environment. LE3H is a crude extract, containing complex compounds with a number of effects. It needs further study through separation to get rid of the low teratogenic properties

The title has been changed to

“Leaf ethanolic extract of Etlingera hemesphaerica Blume mitigates defects in fetal anatomy and endochondral ossification due to mercuric chloride during the post-implantation period in Mus musculus“

14 The article in its present form is not suitable for publication AT ALL. The language certainly needs considerable improvement. The sentences need to be phrased according to the established principles of English grammar Grammatical structure of the manuscript has been double-checked using the Grammarly Insights software (https://app.grammarly.com/)

---

## [Decision Letter · Decision Letter 1]

19 Jan 2021

PONE-D-20-28046R1

Leaf ethanolic extract of Etlingera hemesphaerica Blume mitigates defects in fetal anatomy and endochondral ossification due to mercuric chloride during the post-implantation period in Mus musculus

PLOS ONE

Dear Dr. Ruyani,

Thank you for submitting your manuscript to PLOS ONE. After careful consideration, we feel that it has merit but does not fully meet PLOS ONE’s publication criteria as it currently stands. Therefore, we invite you to submit a revised version of the manuscript that addresses the points raised during the review process.

The revised manuscript has addressed nearly all of the concerns from the reviewers, and they both feel that the manuscript will be acceptable.  However, reviewer 1 would like the details about statistical methods to be added.  This is an important consideration, but one that should be easily remedied.

We look forward to receiving your revised manuscript.

Kind regards,

James A. Marrs

Academic Editor

PLOS ONE

Additional Editor Comments (if provided):

Both reviewers agree that the revised manuscript is a great improvement. Reviewer 1 wants to see that details about statistical methods are included in the manuscript. With this addition, the manuscript should be acceptable. I look forward to seeing this minor revision.

Reviewers' comments:

Reviewer's Responses to Questions

**Comments to the Author**

1. If the authors have adequately addressed your comments raised in a previous round of review and you feel that this manuscript is now acceptable for publication, you may indicate that here to bypass the “Comments to the Author” section, enter your conflict of interest statement in the “Confidential to Editor” section, and submit your "Accept" recommendation.

Reviewer #1: (No Response)

Reviewer #2: All comments have been addressed

2. Is the manuscript technically sound, and do the data support the conclusions?

Reviewer #1: Yes

Reviewer #2: Yes

3. Has the statistical analysis been performed appropriately and rigorously? 

Reviewer #1: Yes

Reviewer #2: Yes

4. Have the authors made all data underlying the findings in their manuscript fully available?

Reviewer #1: Yes

Reviewer #2: Yes

5. Is the manuscript presented in an intelligible fashion and written in standard English?

Reviewer #1: Yes

Reviewer #2: No

6. Review Comments to the Author

Reviewer #1: The authors has addressed most of the comments. One more minor concern: the authors should provide more details In the Statistical Analysis.

Reviewer #2: The authors have undertaken extensive revision. The revised version is much improved and is now suitable for publication in PLoS.

7. PLOS authors have the option to publish the peer review history of their article (what does this mean?). If published, this will include your full peer review and any attached files.

Reviewer #1: No

Reviewer #2: No

---

## [Author Response · Author response to Decision Letter 1]

4 Feb 2021

Editor and Reviewer's note:

Both reviewers agree that the revised manuscript is a great improvement. Reviewer 1 wants to see that details about statistical methods are included in the manuscript. With this addition, the manuscript should be acceptable. I look forward to seeing this minor revision.

Our response:

The data obtained from this study were generalized by the χ2 test of goodness of fit (Table 1) and by multiple comparation, and then the least significant difference (Table 3-6) [29].

---

## [Decision Letter · Decision Letter 2]

8 Feb 2021

Leaf ethanolic extract of Etlingera hemesphaerica Blume mitigates defects in fetal anatomy and endochondral ossification due to mercuric chloride during the post-implantation period in Mus musculus

PONE-D-20-28046R2

Dear Dr. Ruyani,

We’re pleased to inform you that your manuscript has been judged scientifically suitable for publication and will be formally accepted for publication once it meets all outstanding technical requirements.

Kind regards,

James A. Marrs

Academic Editor

PLOS ONE

Additional Editor Comments (optional):

Both reviewers recommended acceptance, and I agree that the manuscript is a strong contribution. Congratulations.

Reviewers' comments:

Reviewer's Responses to Questions

**Comments to the Author**

1. If the authors have adequately addressed your comments raised in a previous round of review and you feel that this manuscript is now acceptable for publication, you may indicate that here to bypass the “Comments to the Author” section, enter your conflict of interest statement in the “Confidential to Editor” section, and submit your "Accept" recommendation.

Reviewer #1: All comments have been addressed

2. Is the manuscript technically sound, and do the data support the conclusions?

Reviewer #1: Yes

3. Has the statistical analysis been performed appropriately and rigorously? 

Reviewer #1: Yes

4. Have the authors made all data underlying the findings in their manuscript fully available?

Reviewer #1: Yes

5. Is the manuscript presented in an intelligible fashion and written in standard English?

Reviewer #1: Yes

6. Review Comments to the Author

Reviewer #1: (No Response)

7. PLOS authors have the option to publish the peer review history of their article (what does this mean?). If published, this will include your full peer review and any attached files.

Reviewer #1: No

---

## [Editor Report · Acceptance letter]

16 Feb 2021

PONE-D-20-28046R2 

Leaf ethanolic extract of *Etlingera hemesphaerica* Blume mitigates defects in fetal anatomy and endochondral ossification due to mercuric chloride during the post-implantation period in *Mus musculus*

Dear Dr. Ruyani:

I'm pleased to inform you that your manuscript has been deemed suitable for publication in PLOS ONE. Congratulations! Your manuscript is now with our production department. 

Kind regards, 

on behalf of

Dr. James A. Marrs 

Academic Editor

PLOS ONE